# Ultra-Low-Voltage-Triggered Silicon Controlled Rectifier ESD Protection Device for 2.5 V Nano Integrated Circuit

**DOI:** 10.3390/nano12234250

**Published:** 2022-11-29

**Authors:** Ruibo Chen, Hao Wei, Hongxia Liu, Zhiwei Liu, Yaolin Chen

**Affiliations:** 1Key Laboratory for Wide Band Gap Semiconductor Materials and Devices of Education, School of Microelectronics, Xidian University, Xi’an 710071, China; 2State Key Laboratory of Electronic Thin Films and Integrated Devices, University of Electronic Science and Technology of China, Chengdu 610056, China

**Keywords:** on-chip ESD, SCR, trigger voltage, overshoot

## Abstract

In this paper, an improved low-voltage-triggered silicon-controlled rectifier (LVTSCR) called an ultra-low-voltage-triggered SCR (ULVTSCR) is proposed and fabricated in a 40-nm CMOS process. By adding an external NMOSs-chain triggering component to the conventional LVTSCR, the proposed ULVTSCR can realize ~2 V lower trigger voltage. Meanwhile, the trigger voltage of the ULVTSCR is adjustable with the number of its incorporated NMOS transistors. Compared with the existing Diodes-chain Triggered SCR (DTSCR) scheme, the NMOSs-chain triggered ULVTSCR possesses a 25% lowered overshoot voltage in the same area consumption, and thus it is more suitable for 2.5 V circuits ESD protections considering the CDM protection applications.

## 1. Introduction

The Silicon-Controlled Rectifier (SCR) has been the most attractive ESD protection component due to its high robustness against ESD stresses [1]. However, the conventional SCR device has a high trigger voltage (Vt1) and a low holding voltage (Vh) [2,3]. Therefore, it cannot provide effective ESD protection in most circuits. To solve these problems, many improved local-based ESD protection schemes were presented, such as Modified Lateral SCR (MLSCR), Low Triggered SCR (LVTSCR), and Diodes-string Triggered SCR (DTSCR) [4,5]. Among them, DTSCR is able to achieve a very low and flexible trigger voltage, and many improved structures based on DTSCR have been proposed in recent years. For example, Chen, Du et al. proposed a novel DTSCR called LTC-DTSCR [6]. By suppressing the trigger of the parasitic SCR of DTSCR, LTC-DTSCR further lowered the trigger voltage. However, the relatively high overshoot voltage and slow turn-on speed of the DTSCR structure impose a hurdle for its usage in the charged device model (CDM) protection [7]. In addition, the DTSCR is not suitable for 2.5 V or above circuits ESD protections because the increased number of the triggering diode will cause large leakage and latch-up risk due to the Darlington effect. As for the LVTSCR, it has the same issues as the conventional SCR: its trigger voltage is too high and hard to adjust to meet the ESD design window of the advanced CMOS process. At this point, several improved LVTSCR structures have been proposed in [8,9], but they focus on improving holding voltage; these devices still have high trigger voltages (~8 V). There are also many new SCR structures proposed. By introducing two gates into SCR, Lin realized a new SCR device with low trigger voltage, low leakage, and low parasitic capacitance [10]. However, it requires an external RC circuit to assist triggering, which will cause huge extra area consumption. P. Galy et al. embedded SCR into BIMOS [11], thus achieving an ultracompact layout, low trigger voltage, and low on-resistance. However, its holding voltage is low, which will increase latch-up risk if the applied voltage domain is relatively high.

In this paper, a new LVTSCR structure with a lower trigger voltage called an ultra-low-voltage-triggered SCR (ULVTSCR) for a 2.5 V voltage domain circuit is first proposed. The new structure incorporates an adjustable NMOSs-chain in the conventional LVTSCR as a driver to its internal embedded NMOS, resulting in a ~25% lower trigger voltage than the existing LVTSCR-based devices. Additionally, ULVTSCR also possesses a high failure current (It2) of 42 mA/μm and a high holding voltage of 3 V, which is latch-up immune for a 2.5 V circuit. Moreover, compared with the DTSCR devices, the ULVTSCR will realize lower overshoot voltage with the same area thanks to its voltage driver triggering component.

## 2. Mechanism

Figure 1 and Figure 2 present the cross-section view and schematic equivalent circuit of the proposed ULVTSCR, respectively. Note that the ULVTSCR is constituted by incorporating an external NMOSs-chain as a voltage driver in a conventional LVTSCR and having the gate of its internal NMOS connected to this driver. Each NMOS in the NMOSs-chain is connected as a “diode connection”, that is, the gate and drain are tied together, the source and sub are tied together, and such units in series form the NMOSs-chain. Additionally, each NMOS in NMOSs-chain is insulated by P+ guard rings and Deep NWell (DNW). The P-substrate of ULVTSCR is connected to the cathode and insulated by a P+ guard ring.

For the conventional LVTSCR, when the ESD stress arrived, the PN junction between the drain and the substrate of the inserted NMOS of the conventional LVTSCR will be the breakdown. Then current flows from the anode to the cathode through the N+ region of the inserted NMOS; thus, a voltage drop is generated on RNW and RPW. Once the voltage drops on RNW and RPW reach the forward bias voltage of the PN junction of P+/PWELL and PWELL/N+, the equivalent transistors Q1 and Q2 will be turned on, then the SCR path shown by the blue curve in Figure 1 is triggered and the LVTSCR begins to discharge the ESD current.

For the ULVTSCR, it has been pointed out before that NMOS in the NMOSs-chain is connected as a “diode connection”, which means MOS in such a connection will behave similar to a diode (being turned on when a positive voltage is applied between the drain and source). So NMOSs-chain formed by such units in series will also behave similar to a diode string but compared to a diode string, NMOSs-chain consumes less area and is free of the Darlington effect. When the ESD stress arrives and reaches the turn-on voltage of the NMOSs-chain, it will be turned on. In addition, the gate of LVTSCR is connected to the NMOSs-chain, as shown in Figure 1 above. So once the NMOSs-chain is turned on, the voltage will be coupled to the gate of the LVTSCR and turn on the inserted NMOS of the LVTSCR. Thereby, current can flow from the anode to the cathode through the inserted NMOS of the LVTSCR, as the path shown by the red line in Figure 1, and charge Q1 and Q2 without avalanche breakdown. The external NMOSs-chain instead avalanche breakdown to assist ULVTSCR in triggering, which enables ULVTSCR to achieve a lower trigger voltage.

## 3. Results and Discussion

In this work, all devices were fabricated in the 40 nm CMOS process because the voltage domain that the 40 nm CMOS process focuses on includes 2.5 V. The quasi-static I-V characteristics of the proposed devices are measured using a Hanwa TED-T5000 transmission line pulsing (TLP) tester with a 10 ns rise time and 100 ns pulse width; the very-fast TLP is measured under a 300 ps rise time with a 5 ns width VFTLP pulse; and the DC leakages are also evaluated.

### 3.1. TLP Results

Figure 3a shows the positive and negative TLP I–V curves of conventional LVTSCR and the proposed ULVTSCR with a 6-NMOSs-chain. The geometrical dimensions of tested LVTSCR and ULVTSCR are presented in Table 1, and the W/L of the thick gate NMOSs in the NMOSs-chain of ULVTSCR is 20 μm/0.3 μm. The conventional LVTSCR features a higher trigger voltage of 8.3 V because LVTSCR is triggered by the avalanche breakdown of its inserted NMOS. While the trigger voltage of an ULVTSCR with a 6-NMOSs string is 6.4 V, which is 1.9 V lower than that of an LVTSCR, compared to an LVTSCR, the trigger of an ULVTSCR includes two stages. Taking ULVTSCR_6 in Figure 3b as an example, ULVTSCR enters the first stage when the voltage reaches Von = 4.2 V due to the conduction of the inserted NMOS and features a slowly rising I–V curve. As the voltage between the anode and cathode increases, the current flow from the anode to the cathode through the inserted NMOS increases until the voltage between the anode and cathode reaches Vtl = 6.4 V and the SCR path is triggered. Then the device enters the second stage and features snap-back characteristics on the I–V curve due to the equivalent resistance of ULVTSCR drops dramatically. As for the negative TLP response, because the NMOSs-chain of ULVTSCR will not be turned during negative TLP, ULVTSCR, and LVTSCR behave the same under negative TLP.

Figure 3b presents the TLP I–V curves of ULVTSCRs, with the number of external NMOSs varying from four to seven. The result shows that the Vtl of ULVTSCR increases from 6.05 V to 6.85 V, with the number of NMOSs in the NMOSs-chain increasing from 4 to 7. This result is expected, because the increased number of NMOSs in the NMOSs-chain will cause the decreased voltage on the gate of the inserted NMOS, leading to the lower current flow through the inserted NMOS, consequently the SCR path has to be triggered by higher Vtl. This result proves that the Vtl of ULVTSCR can be flexibly and coordinately adjusted by adjusting the number of NMOSs in the NMOSs-chain; moreover, it can achieve the same Vtl with fewer NMOSs in series if a higher Vth NMOS is used.

As is shown in Figure 1, D1 represents the length that the drain of the inserted NMOS extends to the NWELL. Figure 4 gives the TLP I–V curves of ULVTSCRs with D1 variation, and the holding voltage Vh and failure current It2 of the tested devices are extracted in Table 2. The test results reveal the trade-off relationship between It2 and Vh. With D1 increasing from 0.25 μm to 2 μm, the It2 decreases from 43 mA/μm to 25 mA/μm, while the Vh rises from 2.1 V to 4.2 V, and the holding current Ih also rises from 2.4 mA/μm to 6.5 mA/μm. Such a result determines that there must be careful optimization on D1 to give consideration to both It2 and Vh. In this work, 0.5 μm is the optimal value of D1, where It2 = 42 mA/μm, Vh = 3 V, and Ih = 4.7 mA/μm. In addition, we can also see from the curves that the impact of D1 on Vtl is negligible.

### 3.2. Overshoot

CDM performance is also very important for an ESD protection device. Figure 5 shows the VFTLP waveforms measured at a current of 1.5 A for the 3-diode string DTSCR and the 6-NMOSs string ULVTSCR. The cross-section of the measured DTSCR is shown in Figure 6. Note that the PWell of the DTSCR is floating. It should be illustrated that the width of NMOSs in the NMOSs-chain and diodes in the diode string are 20 μm and 40 μm, respectively, so the total area of consumption from the 3-string DTSCR and 6-string ULVTSCR is approximately the same.

The test result shows that ULVTSCR possesses CDM performance with a 15 V overshoot, which is 5 V lower than that of DTSCR, despite that the inserted NMOS causes the increased N+ to P+ spacing (Sac) of ULVTSCR. Such a result makes sense. As is reported in work [12,13], the higher the equivalent resistance of the current trigger path of the device, the higher the overshoot. It is clear that the current trigger path of DTSCR is its diode string. As for ULVTSCR, although the NMOSs-chain is the key to the trigger of ULVTSCR, the current trigger path is not the NMOSs-chain but the inserted NMOS (as shown by the red curve in Figure 1). Compared to inserted NMOS, diodes in diode strings are in series, which determines the high turn-on resistance of the diode string. Additionally, the relatively small size of diodes further increases the turn-on resistance of diode strings. If one attempts to compensate for DTSCR’s overshoot by increasing the size of the diode string, then the area cost will be too high. Besides, enlarging the size of diodes cannot increase failure current, which means that for DTSCR, the area consumption on diodes for lowering overshoot is independent, extra area consumption. While enlarging the size of LVTSCR in ULVTSCR can not only lower overshoot but also increase failure current.

### 3.3. Leakage

Leakage current is another critical ESD design metric. Figure 7 shows the DC sweep IV characteristics of the ULVTSCR with a 5, 6, and 7-NMOSs string measured at room temperature; the current was restricted to 100 uA to protect devices under test. The leakage current of these devices under 2.5 V is extracted in Table 3. According to the test result, a conclusion can be drawn that the more NMOSs in the NMOSs-chain, the smaller the leakage. Of course, increasing the length of the NMOSs-chain will increase the trigger voltage Vtl, but a designer can compensate Vtl by bringing the connecting position between the gate of the ULVTSCR and the NMOSs string close to the anode. We can also deduce that the leakage of ULVTSCR is mainly from its NMOSs string, so using NMOS with a higher threshold voltage in the external NMOSs-chain will further decrease leakage.

## 4. Conclusions

In this paper, a new SCR structure based on the LVTSCR was proposed and fabricated in a 40 nm CMOS process. The idea of ULVTSCR is to introduce an external trigger circuit consisting of an NMOSs-chain to turn on its inserted NMOS. The TLP test result showed that ULVTSCR has a trigger voltage below 7 V, lower than the 8.3 V trigger voltage of LVTSCR, and the trigger voltage is easily adjusted by adjusting the number of NMOSs in the NMOSs-chain. Additionally, ULVTSCR also possesses a high failure current (It2) of 42 mA/μm and a high holding voltage of 3 V, which is latch-up immune for a 2.5 V circuit. Thus, the ULVTSCR can be designed to meet the ESD design window of 2.5 V circuits. The V–t waveform of ULVTSCR under CDM pulse with an amplitude of 1.5 A showed that ULVTSCR has an overshoot of 15 V, which is 25% lower than that of DTSCR with an area the same as the measured ULVTSCR. In addition, the leakage current of ULVTSCR under 2.5 V can be restricted to dozens of nA. The above results proved that the proposed ULVTSCR is suitable for the 2.5 V circuits ESD protection applications considering CDM performance.

## Figures and Tables

**Figure 1 nanomaterials-12-04250-f001:**
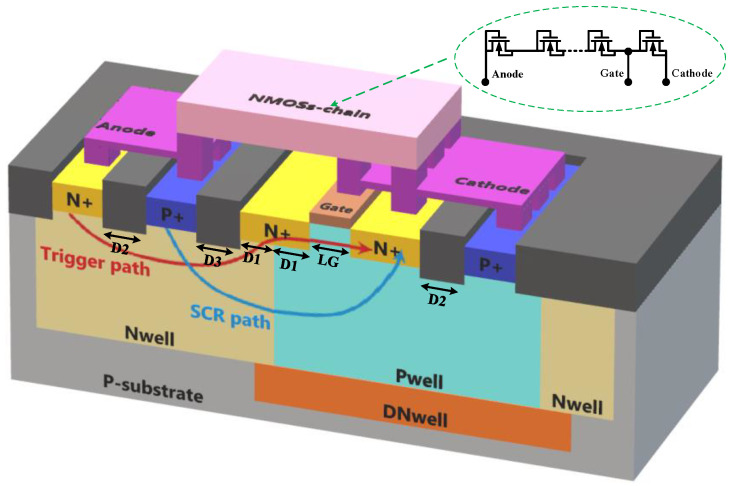
Schematic cross-section of the proposed ULVTSCR device.

**Figure 2 nanomaterials-12-04250-f002:**
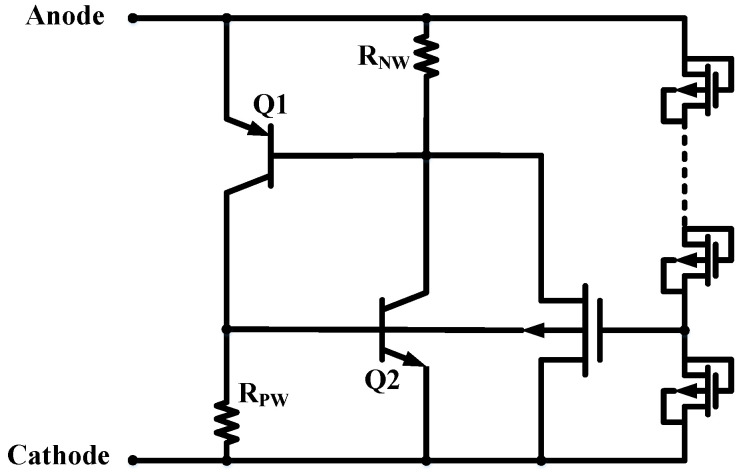
Equivalent circuit of the proposed ULVTSCR device.

**Figure 3 nanomaterials-12-04250-f003:**
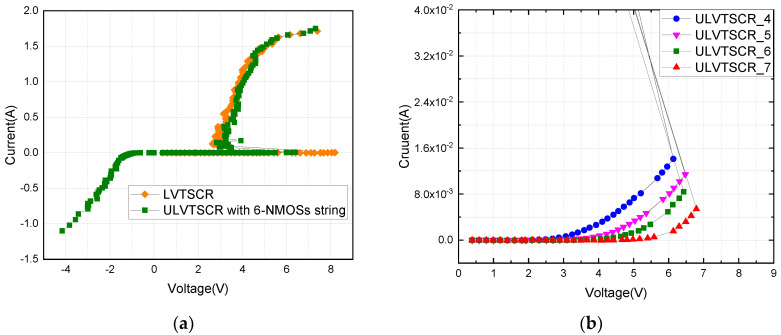
TLP I-V characteristics of: (**a**) 6-NMOSs-chain ULVTSCR and LVTSCR, (**b**) ULVTSCRs with N = 4, 5, 6, and 7.

**Figure 4 nanomaterials-12-04250-f004:**
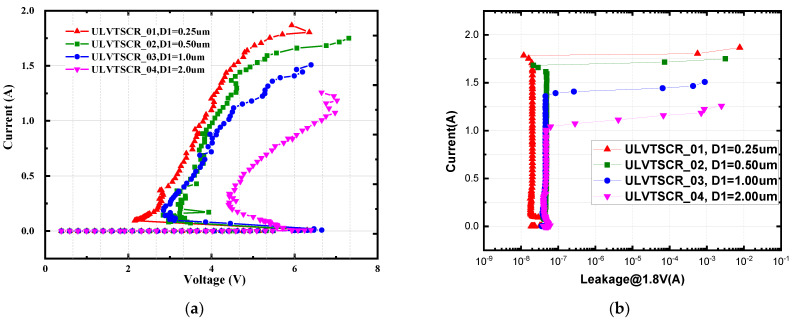
(**a**) TLP I-V characteristics and (**b**) leakage of ULVTSCRs with different D1.

**Figure 5 nanomaterials-12-04250-f005:**
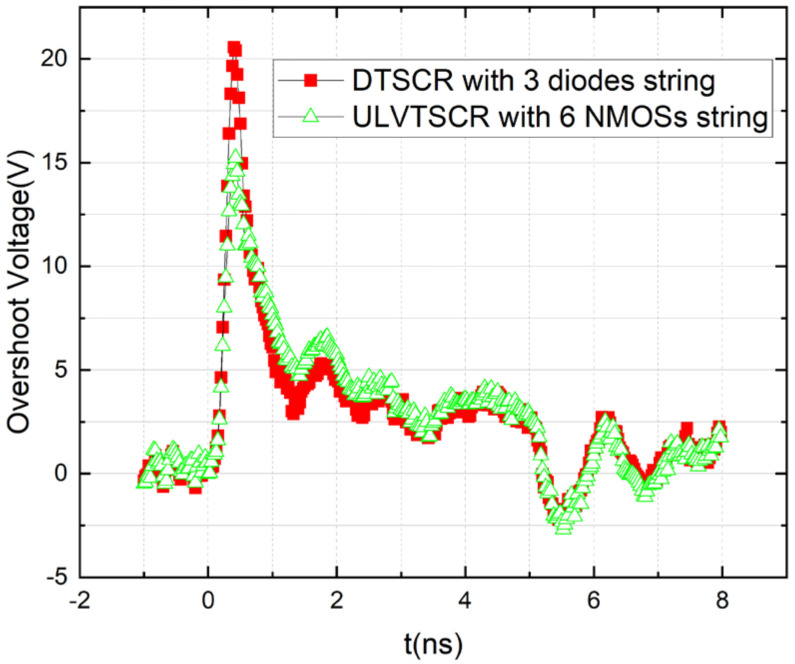
VFTLP waveforms measured at a current of 1.5 A for the 3-string DTSCR and 6-string ULVTSCR.

**Figure 6 nanomaterials-12-04250-f006:**
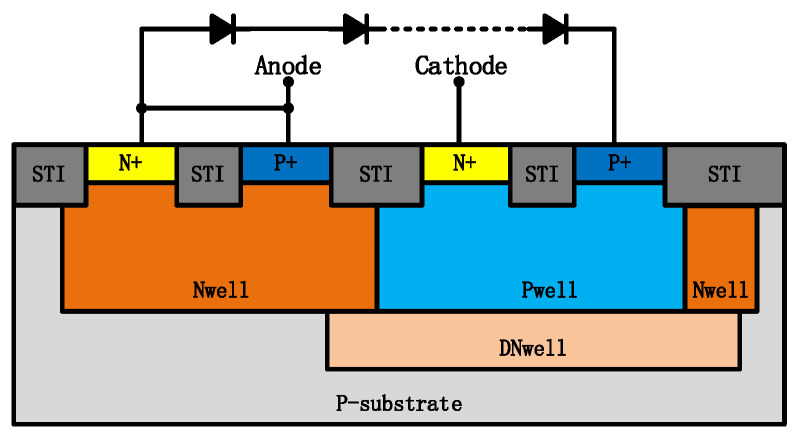
Schematic cross-section of the measured DTSCR.

**Figure 7 nanomaterials-12-04250-f007:**
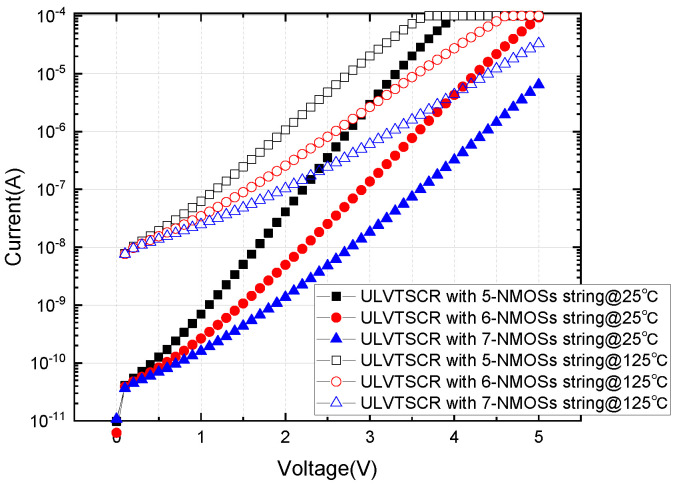
DC sweeps of the ULVTSCR with different NMOSs string. Measured at 25 °C and 125 °C.

**Table 1 nanomaterials-12-04250-t001:** Geometrical dimensions of tested ULVTSCR and LVTSCR.

W(μm)	D1 (μm)	D2 (μm)	D3 (μm)	LG (μm)
40	0.5	2	0.5	0.3

**Table 2 nanomaterials-12-04250-t002:** Extracted Vh and It2 of ULVTSCR with D1 variation.

Device	D1 (μm)	Vh (V)	It2 (A)	It2 (mA/μm)
ULVTSCR-01	0.25	2.1	1.75	43
ULVTSCR-02	0.50	3.0	1.68	42
ULVTSCR-03	1.00	3.1	1.40	35
ULVTSCR-04	2.00	4.2	1.00	25

**Table 3 nanomaterials-12-04250-t003:** Extracted leakage of ULVTSCR with different NMOSs string at 25 °C and 125 °C.

Device	Leakage (nA)	Leakage (nA/μm)
ULVTSCR-05@25 °C	353	8.82
ULVTSCR-06@25 °C	25	0.62
ULVTSCR-07@25 °C	5	0.12
ULVTSCR-05@125 °C	4.78 (μA)	120
ULVTSCR-06@125 °C	811	20.27
ULVTSCR-07@125 °C	241	6.02

## Data Availability

Not applicable.

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
