# Peer review of "Ultra-Low-Voltage-Triggered Silicon Controlled Rectifier ESD Protection Device for 2.5 V Nano Integrated Circuit"

_nanomaterials, 2022, doi:10.3390/nano12234250_

Round 1

Reviewer 1 Report

Journal: Nanomaterials 2022; Manuscript ID: 2030739

 Title : Ultra-low-voltage-triggered Silicon Controlled Rectifier ESD 2 Protection Device for 2.5V Nano Integrated Circuit »

type: paper

Complete List of Authors:

 Ruibo Chen , Hao Wei , Hongxia Liu  , Zhiwei Liu  , and Yaolin Chen

Thanks for Authors for this proposal main focused on ESD protection in 55nm CMOS technology. The proposed study is based on isolated SCR with low voltage trigger circuit with NMOS chain. TLP & VFTLP are used to characterize the silicon test device.

Find hereafter some feedbacks and request to improve the proposal:

-          In introduction, give the ESD robustness in It2/µm and the LU results . add too the voltage range application (or clarify).

-          Please indicate in figures 1;2  some geometrical dimensions and W/L /Thin-thick gate oxide of MOS. What are the values of Rnw/Rpw ?

-          How is biased the Psub? Please indicate the Psub voltage condition in the text?

-          What is the design rule for the N+ overlapping Nwell/Pwell ?

-          Do D1 impacts the overvoltage and the final ESD robustness?

-          What is the footprint of the final protection (area?)

-          Please compare to other solution(s)  for example “Ultracompact ESD Protection With BIMOS Merged Dual Back-to-Back SCR in Hybrid Bulk 28-nm FD-SOI Advanced CMOS Technology ». IEEE Transactions on Electron Devices 2017.

-          Normalize all currents (A/µm) in all results

-          For Vhold point , give the Ihold value . Is it LU compliant ?

-          Fig.4  the inset is too small (impossible to read after print). At which voltage the leakage is monitored ?

-           Fig.5 unities are not clear ?

-          The overvoltage reaches 15V @1.5A (2KV HBM ) thus not fail or latent damage is observed ?

-          The leakage is high @ room temp , what is the result @ 125C?

-          To be rigorous, indicate the negative response/robustness  (equiv. diode)

Mixt 3D TCAD +Spice simulations should be useful for such study, perhaps next time  

Thanks in advance for your effort and correction according to these comments

BR

Reviewer 2 Report

First, the authors have not discussed and linked the present state of the art of the triggered silicon-controlled rectifier ESD protection devices. The introduction does not provide sufficient background and does not include all relevant references. 

It is unclear why the ULVTSCR is constituted by incorporating an external NMOSs-chain as a voltage driver in a conventional LVTSCR and why the 55nm CMOS process has been chosen. What is the advantage of using this process?

Figure 1. Schematic cross-section of the proposed ULVTSCR device - does this figure shows the actual cross-section of chosen 55nm CMOS process? 

Figure 4. TLP I-V characteristics of ULVTSCRs with different D1 - I would suggest separating leakage current to separate figure. 

Figures 5 and 6 have the same description under the picture: VFTLP waveforms measured at a current of 1.5A for the 3-string DTSCR and 6-string ULVTSCR.

Round 2

Reviewer 1 Report

Authors improve their study and take into account about remark ,

thanks for that . 

It is ready for publication in my point of view 

Author Response

Dear reviewer,

Thank you very much for your work on our manuscript. The revision of our manuscript wouldn’t be so smooth without your comments and professional advice. 

It’s our great pleasure to work with you. Thank you and best regards.

Yours sincerely,

Hao Wei

Reviewer 2 Report

I would like to thank the authors for considering all my recommendations and comments, which they duly supplemented and explained in the text. I recommend that in-text citations be listed in ascending numerical order from one to thirteen.

Author Response

Dear reviewer,

We've changed the order of  citations as you suggested. Thank you  very much for your work on our manuscript. The revision of our manuscript wouldn’t be so smooth without your comments and professional advice. 

It’s our great pleasure to work with you. Thank you and best regards.

Yours sincerely,

Hao Wei
